# Tbx1 plays a critical role in focal adhesion dynamics through paxillin regulation

Olimpia Iacolare[1], Rosa Ferrentino[1], Alessandra Altomonte[1], Christopher E Turner[2], Antonio Baldini[3], Daniela Alfano[1]

The T-box transcription factor TBX1 is expressed in the cardio-pharyngeal mesoderm. The correct cell fate decisions of cardio-pharyngeal mesoderm cells are critical, as any defect in this process can alter second heart field morphogenesis and lead to cardiac outflow tract and pharyngeal apparatus defects. The second heart field plays a crucial role in cardiac development by incorporating cardiac progenitors into the heart. It is also the major gene implicated in 22q11.2 deletion (or DiGeorge) syndrome, a primary genetic cause of congenital heart defects associated with hypoplasia of the cardiac outflow tract. The murine model recapitulates the heart phenotype and shows anomalies in the ECM–integrin–focal adhesion pathway. Here, we used a cell culture model to manipulate *Tbx1* levels in order to molecularly and functionally characterize the defective focal adhesions (FAs) caused by *Tbx1* loss and to analyse their dynamics on the ECM. Intriguingly, we found that *Tbx1* regulates FA dynamics by influencing the FA disassembly process. Furthermore, *Tbx1* is required for the paxillin (PXN) signalling pathway and controls cell spreading primarily through *Pxn* regulation. In fact, consistent with this observation, the ectopic expression of PXN rescued the cell spreading and signalling defects caused by *Tbx1* depletion. Finally, our study revealed that, at least in vitro, TBX1 is a critical regulator of cell adhesion by affecting FA turnover.

## Introduction

During the early stages of epithelial morphogenesis, cell–cell and cell–ECM junctions are essential and continually remodelled to accommodate rapid cell proliferation, growth, and differentiation. We and others have highlighted the importance of epithelial architecture and cell adhesion in the second heart field (SHF), particularly in signalling events that control the progenitor cell niche during heart tube elongation (Rana et al, 2014; Francou et al,

2017; Cortes et al, 2018), as well as the development of the outflow tract (OFT), right ventricle, and atria, which contribute to heart morphogenesis (Buckingham et al, 2005). *Tbx1*, a gene encoding a T-box transcription factor, is a key regulator of SHF cardiac progenitors and branchiomeric muscle progenitors of the cardiopharyngeal mesoderm lineage (Diogo et al, 2015), where it sustains cell proliferation and inhibits differentiation (Chen et al, 2009). TBX1 is crucial for the incorporation of cardiac progenitors into the heart and is the major gene implicated in 22q11.2 deletion (or DiGeorge) syndrome, a primary genetic cause of congenital heart defects associated with hypoplasia of the OFT (Lindsay et al, 1999).

Recently, we identified a novel function of *Tbx1* in regulating ECM–cell interactions within the SHF (Alfano et al, 2019). We found that loss of Tbx1 impairs the ECM–integrin–focal adhesion pathway: interfering with this adhesion axis in a mouse embryo culture model using a specific inhibitor causes OFT defects (Alfano et al, 2019). In particular, the specific inhibitor 6-B345TTQ, which blocks the alpha4 integrin–PXN interaction, caused OFT shortening. Published literature suggests that the pharyngeal apparatus (PA) and SHF of *Tbx1* mouse mutants exhibit significant abnormalities in ECM–cell signalling and interactions (Kelly, 2021; Warkala et al, 2021). Moreover, in mice, loss of the ECM–integrin axis leads to several heart defects that are reminiscent of those observed in the absence of *Tbx1* (Mittal et al, 2013; Liang et al, 2014).

Interestingly, bioinformatics analyses of genome-wide target gene data suggested that loss or reduced dosage of *Tbx1* unexpectedly perturbs gene pathways related to focal adhesion dynamics, ECM–receptor interactions, cell movement, and other determinants of cell morphology (Fulcoli et al, 2016; Cirino et al, 2020). These findings provide a link between the transcriptional functions of *Tbx1* and the morphogenetic perturbations observed during mouse development in mutants.

Focal adhesions (FAs) are multifunctional organelles that mediate cell–ECM adhesion, force transmission, cytoskeletal regulation, and signalling. FAs consist of a complex network of trans-plasma membrane integrins and cytoplasmic proteins linking the ECM to the actin cytoskeleton. Paxillin (PXN) is a key component of FAs (Turner et al, 1990), primarily functioning as a molecular scaffold to spatiotemporally

---

[1]Institute of Genetics and Biophysics Adriano Buzzati-Traverso, National Research Council, Naples, Italy   [2]Department of Cell and Developmental Biology, SUNY Upstate Medical University, Syracuse, NY, USA   [3]Department of Molecular Medicine and Medical Biotechnology, University of Naples Federico II, Naples, Italy

Correspondence: daniela.alfano@igb.cnr.it; antonio.baldini@unina.it
Alessandra Altomonte's present address is Centogene GmbH, Rare Disease Company, Rostock, Germany

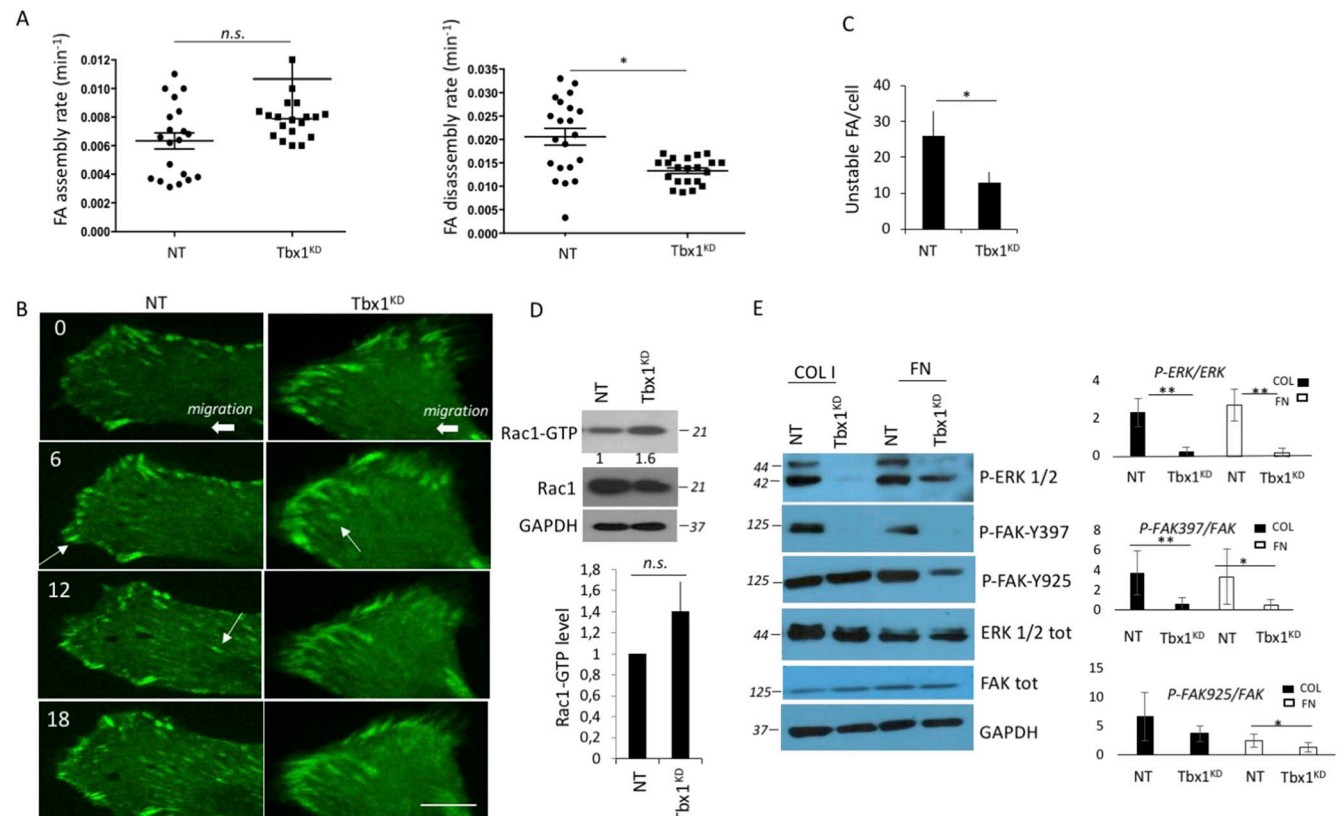

**Figure 1. FA dynamics are altered in *Tbx1*-depleted cells.**
**(A)** Graphs showing the quantification of FA assembly and/or disassembly rate of *Tbx1*-depleted cells (Tbx1[KD]) or control cells (NT). **(B)** Photographs of frames obtained from time-lapse spinning confocal video microscopy movies. Indicated are some of the specific frames that were analysed to assess the number of forming or disassembling FA (arrows) in the lamellipodium of migrating cells, which were transfected with a construct encoding vinculin–GFP fusion protein to mark FAs. **(C)** Graph showing the number of unstable FA/cell normalized for total FA (FA remaining for less than 30 min). **(D)** Total Rac1 activity in collagen-stimulated cells was evaluated by the GST-RBD pull-down assay. The levels of total Rac1 or active Rac1 pulled down by GST-RBD analysed by immunoblotting with anti-Rac1 antibody. GAPDH was used as a loading control. **(E)** ECM-stimulated cells were analysed for the P-FAK (residues Y397 or Y925) and P-ERK1/2 levels by immunoblotting with specific antibody. GAPDH was used as a loading control. Graphs represent quantitative densitometric analysis from at least three experiments (right panels). N = 3 biological replicates. The scale bar represents 10 μm. *P < 0.05, **P < 0.01.

integrate diverse signalling networks, transducing and coordinating dynamic intracellular responses to a variety of stimuli. Through its interactome, paxillin has been shown to regulate FA growth, stabilization, and disassembly to enable migration on 2D surfaces, as well as invasion through 3D-ECM (Turner, 2000a; Alpha et al, 2020).

Here, we used cultured cells to manipulate *Tbx1* and *Pxn* levels in order to molecularly and functionally characterize the defective FAs caused by Tbx1 loss, analyse their dynamics on the ECM, and investigate whether paxillin loss per se is sufficient to recapitulate FA anomalies caused by *Tbx1* depletion.

## Results

### Tbx1 is critical for focal adhesion dynamics

*Tbx1* loss led to FA defects (Alfano et al, 2019). The difference between control (NT) and *Tbx1* knockdown (Tbx1[KD]) cells, in terms of the number and size of FAs, prompted us to investigate

whether FA dynamics was influenced in *Tbx1* loss-of-function conditions. In this study, we used an undifferentiated myoblast cellular model (C2C12 cells), given that cardio-pharyngeal mesoderm develops into a broad range of pharyngeal structures and cell types encompassing musculoskeletal and connective tissues, giving rise to the craniofacial muscles and the heart. Live-cell imaging of C2C12 cells, transiently expressing vinculin–GFP, revealed that although the assembly rates of FAs were similar, there was a significant reduction in disassembly rates between *Tbx1*[KD] and control cells (Figs 1A and S1A). Indeed, FAs in control cells were notably more dynamic than those of Tbx1[KD] cells (Fig 1B and C; Video 1 and Video 2). During the interval of observation, FAs in Tbx1[KD] cells were often static, whereas in control cells, many of them underwent bouts of formation and disassembly. Analysis of individual frames from the movies showed that the focal adhesion disassembly rate was severely impaired: the number of unstable FAs, that is, with a lifetime of less than 30 min, was reduced by nearly 50% (Fig 1B and C). We previously indicated that Tbx1[KD] cells, like control cells, exhibited bundled microtubule filaments that associate

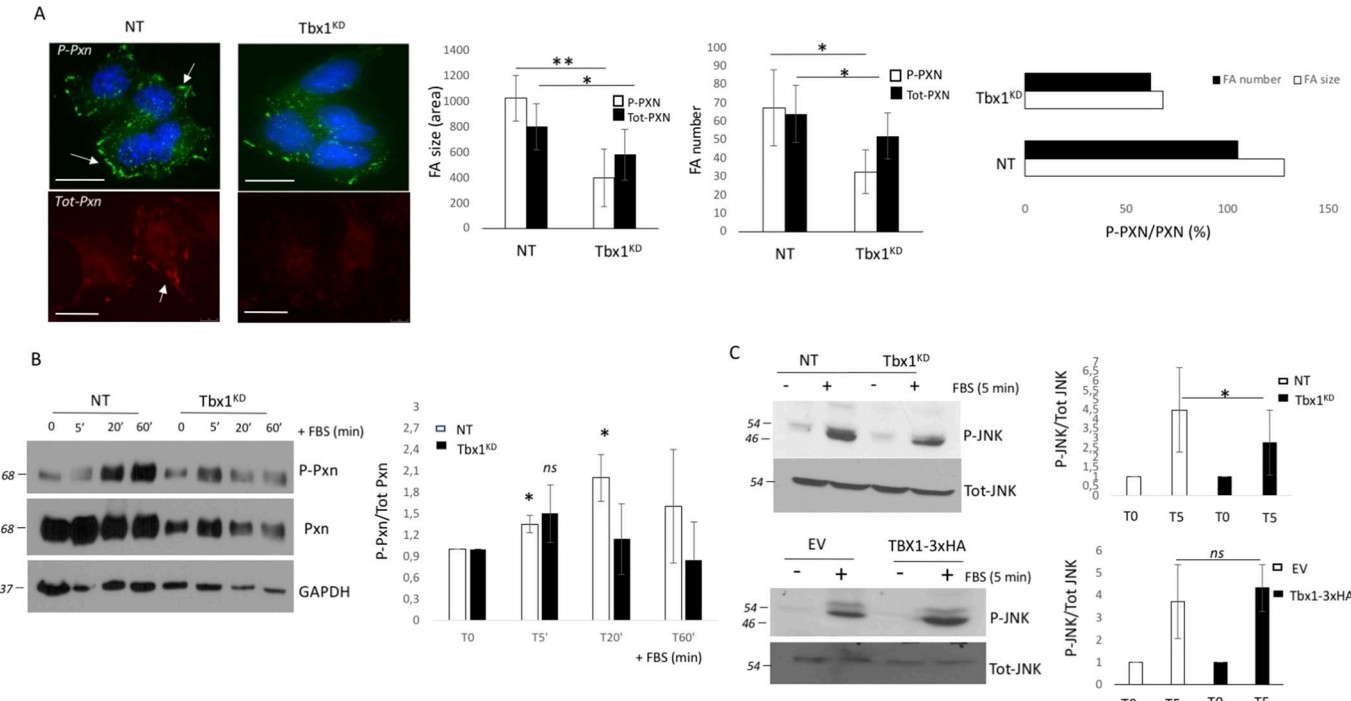

**Figure 2.** *Tbx1* loss causes a defect in the Pxn-mediated molecular signalling pathway.
**(A)** P-Pxn (P-Tyr118) or tot-Pxn immunostaining (see arrows) is decreased in *Tbx1*-deleted cells compared with the control cell. Focal adhesion (FA) analysis, including FA size and number, was performed on cells stained with either anti-phospho-paxillin (P-Pxn) or anti-total paxillin (Tot-Pxn) antibodies and analysed using ImageJ software. **(B)** Cells were serum-starved for 24 h, stimulated with 10% serum, and lysed at the indicated time points. Cell lysates were analysed by immunoblotting with anti-P-Pxn (P-Tyr118) or total Pxn. GAPDH was used as a loading control. N = 3 biological replicates. The graph shows the relative densitometric analysis. **(C)** *Tbx1*-depleted or TBX1-overexpressing cells were stimulated with serum and subjected to immunoblotting analysis with specific P-JNK or anti-Tot-JNK antibodies. The graphs show the relative densitometric analysis (scale bars: 25 μm). *P < 0.05, **P < 0.01 compared with NT.

with peripheral FAs (Alfano et al, 2019), suggesting that the FA turnover defect is probably not linked to microtubule dynamics. Our previous results showed that P-Pxn–marked FAs in growth factor–stimulated cells were fewer and smaller, as compared to control cells (Alfano et al, 2019). P-Pxn is known to occupy the outer/nascent FAs, and it is localized to dynamic adhesions (Ballestrem et al, 2006), with previous data showing Pxn involvement in FA disassembly (Webb et al, 2004; Zehrbach et al, 2023). These findings are consistent with an FA turnover defect and support the role of TBX1 in regulating FA dynamics.

Small GTPases of the Rho family are known to play a pivotal role in regulating FA dynamics (Jaffe & Hall, 2005), acting as regulatory convergence nodes that dictate cytoskeletal and adhesion assembly and organization (Ridley & Hall, 1992). Paxillin coordinates the spatiotemporal activation of Cdc42, Rac1, and RhoA GTPases by recruiting GEFs and GAPs along with specific effector proteins to FAs, thus regulating cell adhesion and spreading during polarized cell migration (Ridley, 2015). In fact, Pxn interacts directly with FAK kinases, and also possibly with β1 integrin (Turner & Miller, 1994; Nikolopoulos & Turner, 2000; Brown & Turner, 2004), and also plays a key role in coordinating cell–ECM signalling (Turner et al, 1990), to regulate cytoskeleton reorganization, particularly via coordination of Rho GTPase family activity (Turner, 2000b; Brown & Turner, 2004; Deakin & Turner, 2008). Therefore,

we analysed the Rho family GTPase activation status of cells plated onto collagen I (COLI), and found that the active Rac1 level increased in Tbx1[KD] cells compared with the control (Fig 1D).

In *Tbx1* loss-of-function conditions, FAK activation (at both phosphorylated Y397 and Y925 residues) was impaired, also indicative of an FA turnover defect; moreover, also the downstream MAPK pathway was defective when cells were plated on ECM (Fig 1E).

### Tbx1 positively regulates paxillin signalling

The function and localization of Pxn are tightly regulated by phosphorylation, which may be induced by both integrin-dependent adhesion to ECM and growth factor stimulation (Burridge et al, 1992; Bellis et al, 1997; Schaller & Schaefer, 2001). Accordingly, we previously found that *Tbx1* depletion caused Pxn phosphorylation impairment when cells were plated on ECM, but not on plastic, indicating that *Tbx1*-depleted cells had a defective outside-in signalling (Alfano et al, 2019). To understand whether Pxn activation by stimulation with growth factors is also affected by Tbx1 depletion, we assayed P-Tyr118 (P-Pxn) after growth factor induction in control and Tbx1[KD] cells by performing an immunofluorescence experiment (Fig 2A). Although *Tbx1* knockdown cells show a reduction in focal adhesions—both in size and

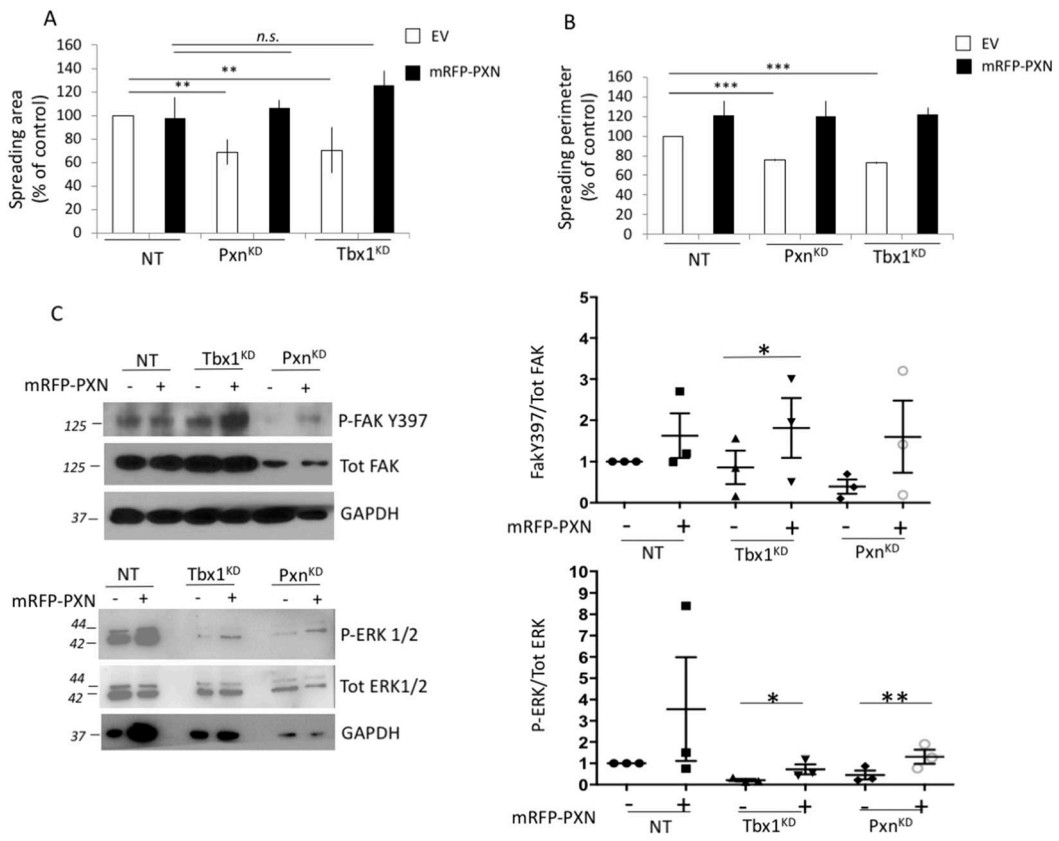

**Figure 3.  Ectopic PXN rescues spreading defect of *Tbx1*-depleted cells.**
*Tbx1* knockdown cells (Tbx1[KD]) were transfected with a construct encoding mRFP-Pxn fusion protein or with an empty vector. **(A, B)** These cells were tested for their ability to spread on collagen I; cell spreading area (A) and spreading perimeter (B) of 70 cells were evaluated. N = 3 biological replicates. *Pxn*-depleted (Pxn[KD]) cells were used as a control for spreading. **(C)** Pxn-associated kinase activation was assessed in *Tbx1*-depleted cells expressing or not ectopic PXN. Cell lysates were analysed by immunoblotting with antibodies to the indicated phosphorylated proteins. Phosphorylated FAK (P-FAK) and ERK1/2 (P-ERK1/2) levels were normalized respectively to total FAK and ERK1/2 levels and quantified by densitometric scanning (right panels). GAPDH was used as a loading control. The values are the means ± s.e.m. of three experiments. *$P < 0.05$, **$P < 0.01$, ***$P < 0.001$ compared with NT.

in number—compared with control NT cells, when analysed with an anti-total paxillin antibody, this decrease is even more pronounced when using an antibody specific for the phosphorylated form of paxillin. These findings suggest that in addition to the down-regulation of paxillin expression, there is also an impairment in the downstream signalling pathway.

Accordingly, Western blot analysis showed that in control cells, P-Pxn increased after 20 min of FBS induction and was still maintained after 60 min (Fig 2B). In contrast, in Tbx1[KD] cells, P-Pxn was reduced, with only a small, non-significant increase in signal after 5 min of FBS induction, and no increase at later time points, showing that *Tbx1* depletion prevented FBS-induced Pxn activation. Previous studies indicate that Pxn is an important mediator also of JNK signalling in adhesion dynamics (Huang et al, 2003); therefore, we also tested the activation of the JNK kinase in response to FBS stimulation in Tbx1[KD] or Tbx1-overexpressing cells and found that JNK activation seems to be correlated to the *Tbx1* dosage (Fig 2C). Taken together, these data point to a role of TBX1 in regulating the Pxn signalling cascade.

## Exogenous *Pxn* rescues the cell spreading and signalling defects caused by *Tbx1* depletion

*Tbx1* depletion causes defective cell spreading and signalling defects (Alfano et al, 2019). We asked whether these defects may be due to *Pxn* gene down-regulation, given that *Pxn* is a target of *Tbx1*. We expressed a mRFP-Pxn fusion protein to perform a rescue experiment in C2C12 cells (Fig S1B). Quantitation of cell spreading, measured by the cell area and perimeter of fluorescent cells, showed that the exogenous expression of Pxn rescued the spreading defect of both *Tbx1*- and *Pxn*-depleted cells (Tbx1[KD] and Pxn[KD], respectively) (Figs 3A and B and S1C). Pxn overexpression had no effects on spreading of control cells, possibly because C2C12 cells express high levels of endogenous Pxn; thus, a small dosage increase is not likely to have an impact. In addition, we also explored the signalling pathway affected in Tbx1[KD] cells. In particular, exogenous *Pxn* partially rescued ECM-mediated phosphorylation of FAK (Y397) and P-ERK1/2 in Tbx1[KD] cells (Figs 3C and S2A). Taken together, these data point to a role of TBX1 in regulating cell shape and adhesion in a Pxn-dependent manner.

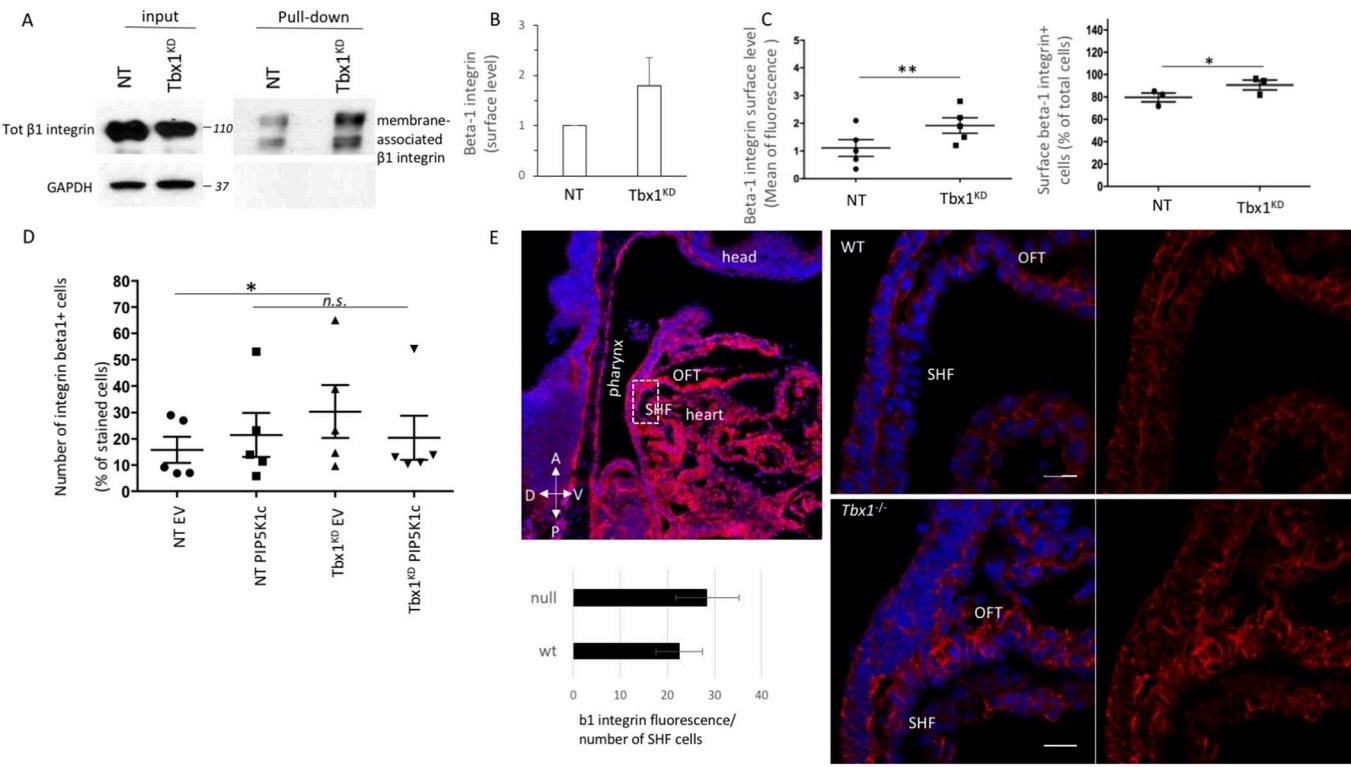

**Figure 4.** *Tbx1* **loss impacts the integrin trafficking.**
**(A)** Total lysates (inputs) and agarose streptavidin bead eluates (pull-down) were subjected to electrophoresis and subsequently probed with the anti-beta1 integrin antibody. GAPDH was used as a loading control. **(A, B)** Membrane-associated beta1 integrin levels of pull-down fractions in (A) were quantified by densitometric scanning. **(C)** Cells were plated on COLI-coated dishes and stained for surface levels of $\beta$1 integrin, then analysed by flow cytometry. Values are the mean fluorescence of the population $\pm$ s.e.m. or the mean of a number of surface beta1 integrin–positive cells (expressed as % of total cells) of n = 5 biological replicates. **(D)** Cells transiently transfected with a PIP5K1c-encoding plasmid were plated on COLI-coated dishes and stained for surface levels of $\beta$1 integrin, then analysed by flow cytometry. Values are the mean of a number of surface beta1 integrin–positive cells $\pm$ s.e.m. (expressed as % of total cells). n = 5 biological replicates. *$P$ < 0.05. **(E)** Sagittal sections of E9.5 embryos showing the distribution of active form of integrin beta1 (clone 9EG7) in the second heart field level of WT and *Tbx1$^{-/-}$* mutant embryos. The first panel is a low-magnification view in which the white box indicates the location of higher magnification images (scale bar: 20 $\mu$m). n = 4. A, anterior; P, posterior; V, ventral; D, dorsal. *$P$ < 0.05, **$P$ < 0.01.

## FA turnover defect is associated with integrin trafficking anomalies

The dynamic assembly and disassembly of integrin–ECM adhesions are crucial for cells to adhere to and detach from the ECM during cell migration, with integrin recycling being central to the regulation of this process of dynamic adhesion turnover (Gardel et al, 2010; Vicente-Manzanares & Horwitz, 2011). In particular, collagen receptor integrins recycle between the plasma membrane and endosomes, facilitating the formation and turnover of FA. Therefore, we investigated the levels of membrane-associated integrin beta1 when cells were plated on COLI. Previous data from in vitro and in vivo systems indicated that Tbx1 loss led to down-regulation of the beta1 integrin subunit (Alfano et al, 2019; Fig S1D). Consistently, we found that *Tbx1*-depleted cells had decreased levels of total beta1 integrin (Fig 4A). However, when we assessed the amount of membrane-associated integrin by labelling surface proteins with Sulfo-NHS-SS-biotin and performing subsequent pull-down of the labelled protein with streptavidin beads, we observed a slight increase in the membrane-associated beta1 integrin fraction in *Tbx1*-silenced cells compared with control cells (Fig 4B). The

absence of the internal control protein GAPDH in the pull-down lanes after Western blotting indicated the quality of the pull-down (Fig 4A).

Furthermore, cytofluorimetric analysis of membrane-associated beta1 integrin showed that ECM-stimulated cells, stained with anti-beta1 antibody, revealed changes in integrin distribution (Figs 4C and S1E). Accordingly, in vivo, where cells are attached to the ECM, *Tbx1$^{-/-}$* embryos exhibited increased active beta1 immunostaining in the SHF and OFT regions compared with WT embryos (Fig 4E).

These results suggest that the FA disassembly defect caused by *Tbx1* loss is associated with integrin trafficking anomalies, particularly a defective internalization of focal adhesion components. To further investigate this, we tested whether membrane-associated integrin beta1 levels varied in cells overexpressing PIP5K1c kinase. We know that Tbx1 potentially down-regulates PIP5K1c at the transcriptional level (Fulcoli et al, 2016; Cirino et al, 2020) and that in Tbx1$^{KD}$ cells, PIP5K1c is decreased (Figs 4D and S2B), and PIP5K1c is involved in receptor endocytosis and exocytosis, including the endocytosis of integrins (Di Paolo et al, 2002; Ling et al, 2002; Thapa et al, 2012). Interestingly, in Tbx1$^{KD}$ cells, the

overexpression of the PIP5K1c kinase led to a decrease in membrane-associated beta1 integrin levels, similar to the levels observed in control cells (Fig 2), suggesting that integrin beta1 recycling may be impaired in the Tbx1[KD] cells. To investigate the mechanistic insight underlying the reduced integrin surface expression in Tbx1[KD] cells, we examined talin expression levels and found a small increase in Tbx1[KD] cells compared with control cells (Fig S2C).

Further studies are needed to molecularly investigate which integrin trafficking pathways—between recycling and endocytosis—are altered in Tbx1 loss-of-function cells.

## Discussion

We initiated this work in search of molecular and functional characterization of the adhesion defects associated with the loss of function of the TBX1 transcription factor. Previously, we found that the ECM–integrin–focal adhesion (FA) pathway is altered in the SHF of *Tbx1* mutants and that the integrity of the ECM–integrin–FA axis is essential for OFT morphogenesis (Alfano et al, 2019). To better understand the molecular mechanisms underlying the adhesive anomalies observed in *Tbx1* gene loss-of-function mice, we used a cellular system that allows manipulation of *Tbx1* levels. We explored the consequent changes in cell morphology and adhesive properties, monitoring FA dynamics at the single-cell level. Interestingly, we found that *Tbx1* loss impacts FA turnover, particularly affecting their disassembly rate. The spreading defect, as well as the downstream signalling involved, was rescued by the ectopic expression of paxillin (PXN), suggesting that the defect in *Tbx1*-depleted cells is primarily due to the loss of *Pxn*. In addition, both ECM- and growth factor–mediated Pxn signalling were impaired in these cells. This work uncovers a novel function of TBX1 as a modulator of FA turnover. Although the relationship between Rac1 activity and FA dynamics remains somewhat unclear (Webb et al, 2004; Deakin et al, 2012; Steffen et al, 2013), it is plausible to speculate that probably Rac1 may not be essential for overall FA dynamics. Instead, it might specifically influence the dynamics of individual protein components of FA. Indeed, Rac1 activation downstream of TBX1 appears to be spatially restricted, potentially exerting localized effects on FA behaviour, as global levels of GTP-bound Rac1 were not significantly altered, suggesting a possible Rac1 dysregulation (West et al, 2001). Dynamic remodelling of adhesions, through rapid endocytic and exocytic trafficking of integrin receptors, is a key mechanism employed by cells to regulate integrin–ECM interactions, and thus cellular signalling, during processes such as cell migration. Endocytic trafficking of integrins provides an important complementary mechanism for regulating integrin–ECM adhesion turnover (Caswell & Norman, 2006; Caswell et al, 2009; Scita & Di Fiore, 2010). We found that *Tbx1* loss led to an increase in integrin beta1 levels on the cell surface, suggesting that Tbx1 may act as a regulator of integrin trafficking. Further studies are needed to determine whether the altered surface levels of integrin beta1 correlate with changes in its activation status.

Accordingly, bioinformatics analyses of genome-wide target gene data indicated that loss or reduced dosage of *Tbx1* unexpectedly perturbs gene pathways related to FA dynamics, ECM–receptor interactions, cell movement, and other determinants of cell morphology (Fulcoli et al, 2016; Cirino et al, 2020). In particular, *Tbx1* loss down-regulates genes involved in integrin turnover, including those encoding dynamin2, PIP5KIgamma (*PIP5K1c*), Rab family members, and Arf6 (Fulcoli et al, 2016; Cirino et al, 2020). Published data suggest that FA disassembly anomalies are associated with PIP5KI production in response to integrin–ECM adhesion (Di Paolo et al, 2002; Ling et al, 2002); moreover, PIP5KIgamma mediates the endocytosis of active beta1 integrin and drives FA disassembly (Thapa et al, 2012). Further investigations are needed to explore the molecular mechanisms underlying FA turnover imbalances caused by low *Tbx1* dosage and to dissect the endocytic pathways, distinguishing clathrin- and caveolin-mediated routes, and determining whether they follow short-loop or long-loop pathways.

## Materials and Methods

### Mouse lines

The mouse lines $Tbx1^{lacZ/+}$ (null allele, here referred to as $Tbx1^{-/+}$) (Lindsay et al, 2001) were maintained in a clean facility in a C57BL/6N background. The mouse line used here is available through the following public repository: $Tbx1^{Lacz}$, EMMA repository EM:02137. Genotyping was carried out according to instructions provided by the original reports. The developmental stage of embryos was evaluated by considering the morning of vaginal plug as E0.5 and by counting somites of embryos. Embryos were collected at E8.5 and E9.5. Animal studies were carried out according to the animal protocol 257/2015-PR (licensed to the AB laboratory) reviewed by the Italian Istituto Superiore di Sanità and approved by the Italian Ministero della Salute, according to Italian regulations.

### Cell culture and transfections

Mouse C2C12 undifferentiated myoblast cells obtained from the ATCC (catalogue #CRL-1772) were cultured in DMEM (Invitrogen) supplemented with 10% FBS and L-glutamine (300 $\mu g$/ml) and were free of *Mycoplasma*. Primary MEFs were isolated from $Tbx1^{+/-}$, $Tbx1^{-/-}$, and WT embryos at E13.5. To this end, the internal organs, head, tail, and limbs were removed. Cells were cultured in DMEM with 20% FBS and 1% NEAA and used for a maximum of three passages. Cells were incubated at 37°C in 5% $CO_2$. For siRNA transfection, cells were seeded at $1.2 \times 10^5$ per well in six-well plates and transfected with a pool of Silencer Select Pre-Designed *Tbx1* siRNA or *Pxn* siRNA (Life Technology; final concentration, 50 nmol/litre) in antibiotic-free medium using Lipofectamine RNAiMAX Reagent (Life Technology) according to the manufacturer's instructions. 24 h later, cells were transfected with a construct encoding TBX1-3xHA, mRFP-PXN, Vnc-GFP, PIP5K1c, or their relative empty vectors. Cells were collected and processed for further analysis 48 h after siRNA transfection.

## Immunoblotting

Cells were harvested in lysis buffer (50 mM Tris–HCl, pH 7.6, 2 mM EDTA, 150 mM NaCl, 0.5% Triton X-100) supplemented with 1 mM phenylmethylsulphonyl fluoride and 1× cOmplete mini EDTA-free protease inhibitor and PhosSTOP phosphatase inhibitor (Roche). Debris was removed by centrifugation at 10,000$g$ for 20 min at 4°C, and the protein content was assessed by a Bradford protein assay. Nuclear proteins were extracted as described previously (Fulcoli et al, 2016). Proteins were separated by SDS–PAGE and transferred to Immobilon-P PVDF membranes (Bio-Rad). Membranes were subsequently incubated for 1 h at RT in TBST buffer (125 mM Tris–HCl [pH 8.0], 625 mM NaCl, 0.1% Tween-20) containing 5% BSA and further incubated at 4°C for 16 h with the following primary antibodies: phospho-Pxn (#2541; Cell Signaling), phospho-FAK (Y397) (#8556; Cell Signaling), phospho-FAK (Y925) (#sc-11766; Santa Cruz), phospho-ERK (p44/42) (#9101; Cell Signaling), Pxn (ab32084; Abcam), FAK (#1688; Santa Cruz), ERK (#9102; Cell Signaling), Tbx1 (ab18530; Abcam), PIP5K1c (ab109192; Abcam), talin (SAB4200694; Sigma-Aldrich), GAPDH (ab125247; Abcam), lamin B1 (ab133741; Abcam). Secondary HRP-conjugated mouse and rabbit (GE Healthcare) or rat (Dako) antibodies were used at a dilution of 1:5,000. Membranes were developed using the enhanced chemiluminescence system (GE Healthcare). X-ray films were scanned and processed using ImageJ software for densitometric analysis.

## Rac1 activity assay

The pull-down assay to measure Rac1 activity was performed using a Rac activation assay kit, according to the manufacturer's protocol (#17-441; Millipore). Subconfluent C2C12 cells were washed in serum-free medium and plated onto COLI-coated dishes for 20 min at 37°C before the addition of lysis buffer (150 mM Tris–HCl, pH 7.5, 1 mM EDTA, 500 mM NaCl, 10 mM MgCl$_2$, 1% Triton X-100, 0.5% sodium deoxycholate, 0.1% SDS, 10% glycerol, 0.5% 2-mercaptoethanol) at 4°C. A protein quantitation assay (Bio-Rad) was performed, and equal amounts of total protein were used for each pull-down assay. Lysates were incubated with GST–PAK1 PBD on glutathione–agarose beads for 1 h at 4°C, and then, the beads were washed four times in lysis buffer. The agarose beads were boiled in SDS–PAGE sample buffer to release active Rac1 proteins, which were then processed for immunoblotting with an anti-Rac1 antibody. The total cell lysate (30 $\mu$g) per sample was used to detect total Rac1.

## Cell spreading

Flat-bottom 96-well microtitre plates were coated with 10 $\mu$g/ml collagen I (Sigma-Aldrich) or 1% heat-denatured BSA in PBS (uncoated plastic) as a negative control, and incubated overnight at 4°C. Plates were then blocked for 1 h at RT with 1% heat-denatured BSA in PBS. Cells were harvested using trypsin and allowed to recover for 1 h at 37°C, then washed three times in PBS. 10$^5$ cells were allowed to spread on the substrate for 20 min and fixed with PFA without washing. Cells were then stained for F-actin, and area and perimeter quantified with ImageJ software.

## FA size analysis and spinning disc live-cell microscopy

C2C12 cells were transfected with siRNAs targeting *Tbx1* or with non-targeting control siRNA; 24 h later, cells were transiently transfected with Vnc–GFP. Live-cell imaging was carried out at 48 h post-siRNA treatment as described above for 2–3 h to determine focal adhesion assembly and disassembly rates under steady-state conditions. Cells were seeded on glass-bottomed dishes (35-mm round dishes, MatTek). Confocal images were acquired with a Nikon spinning disc inverted confocal microscope (Nikon Eclipse Ti). The images were processed for estimation of various parameters using Focal Adhesion Analysis Server (FAAS) (Berginski et al, 2011). This software does not distinguish FAs from the smaller focal complexes, so we refer to all GFP structures larger than 0.05 $\mu$m$^2$ (2 pixels) as FAs. Kinetics of FA assembly and disassembly were performed as previously described with minor modifications (Webb et al, 2004). The rate constants for FA assembly and disassembly were obtained by calculating the fluorescent intensity of individual FAs as a function of time.

## Surface biotinylation assay

After 72 h of siRNA transfection, cells were plated on COLI-coated dishes for 20 min at 37°C and then washed three times in cold PBS surface-labelled at 4°C with 0.3 mg ml$^{-1}$ NHS-LC-biotin (Thermo Fisher Scientific) in PBS for 2 h on ice. Labelled cells were washed four times in cold PBS, and excess of biotin was removed by incubation with RPMI/10% FCS for 20 min at 4°C. After three PBS washes, cells were lysed in 50 mM Tris, pH 8, 150 mM NaCl, 5 mM EDTA, 1% NP-40, 25 mM NaF, 2 mM Na$_3$VO$_4$, phosphatase inhibitor cocktail (Calbiochem), and protease inhibitor cocktail (Roche). Lysates were sonicated, clarified by centrifugation at 16,000$g$ for 10 min, and incubated with streptavidin agarose beads (Thermo Fisher Scientific) for 2 h at 4°C in agitation. Beads were washed four times with lysis buffer, and pulled-down $\beta$1 integrins were analysed by SDS–PAGE and Western blotting.

## Flow cytometry analysis

For analysis of surface beta1 integrin and/or surface-active $\beta$1 integrin levels, cells were scraped into PBS, fixed in 4% PFA–PBS for 30 min on ice, and blocked in 5% BSA–PBS for 30 min on ice. Cells were then incubated with monoclonal 10 $\mu$g/ml anti-integrin $\beta$1 (A-4 sc-374429; Santa Cruz) overnight at 4°C. Finally, the cells were washed and analysed by flow cytometry using a FACScan (Becton Dickinson).

## Histology and immunofluorescence

E9.5 (23–25 somites) mouse embryos were fixed in 4% PFA and embedded in OCT. 10-$\mu$m sagittal or transverse cryosections were subjected to immunofluorescence using anti-integrin beta1 (clone 9EG7) primary antibody (10 $\mu$g/ml) and incubated with the appropriate secondary antibody labelled with a fluorescent probe (for 1 h; dilution 1:400). C2C12 transfected cells were fixed in 4% PFA and incubated with anti-phospho-Pxn (Abcam) or anti-Tot-Pxn (Abcam) antibodies (2 $\mu$g/ml).

### RNA extraction, cDNA synthesis, and quantitative RT–PCR

RNA was extracted from E9.5 embryos using TRI Reagent (Ambion/Applied Biosystems) according to the manufacturer's protocol. Extracted RNA was treated with DNA-free Kit (Ambion/Applied Biosystems). cDNA was synthesized from 1 $\mu$g total RNA (normalized via UV spectroscopy) using the High-Capacity cDNA Reverse Transcription Kit, according to the manufacturer's instructions (Applied Biosystems). Target cDNA levels were compared by qRT–PCR in 20-$\mu$l reactions containing 1× SYBR Green (FastStart Universal SYBR Green Master [Rox], Roche), 0.2 $\mu$mol/litre of each primer. Primers were as follows: for *Tbx1* amplification, forward primer 5′-CTGACCAATAACCTGCTGGATGA-3′ and reverse primer 5′-GGCTGATATCTGTGCATGGAGTT-3′; for *Itgb1* amplification, forward primer 5′-GGTTTGTCATGGACATGCTG-3′ and reverse primer 5′-TGGGGCCTAGTAACACCAAG-3′; for *Rpl13a* amplification, forward primer 5′-CCCTCCACCCTATGACAAGA-3′ and reverse primer 5′-CTGCCTGTTTCCGTAACCTC-3′. Results were normalized against *Rpl13a* and compared by relative expression and the delta–delta cycle threshold method for fold change calculations with StepOne v2.3 software (Applied Biosystems).

### Proliferation assay

For the cell proliferation assay, C2C12 cells were plated at $1.5 \times 10^4$ cells/cm$^2$ and cell viability was measured at 16 h, 24 h, or 36 h, using the colorimetric CyQUANT-GR cell proliferation assay (Invitrogen), following the manufacturer's instructions. Briefly, triplicate samples were washed with PBS and stored at –80°C for 2 h. After thawing, cells were incubated with a mix of cell lysis buffer and CyQUANT-GR dye. Absorbance was analysed at 480–520 nm, using Fluoroskan Ascent FL Microplate Fluorometer and Luminometer (Thermo Fisher Scientific).

### Statistical analysis

GraphPad Prism (GraphPad Inc.) software was used. Statistical analysis was carried out where indicated using data from three or more independent experiments each in triplicate, unless stated otherwise. Differences between data sets were determined using an unpaired *t* test. Significant differences were considered at $P < 0.05$.

## Data Availability

All data produced or examined during this investigation are comprehensively presented in this published article or in the Supplementary Material. The source data and any additional information in this article will be shared upon request.

## Supplementary Information

## Acknowledgements

We are grateful to Maddy Parsons for providing the constructs encoding mRFP-PXN and Vnc-GFP, and Antonella De Matteis and Leopoldo Staiano for providing the plasmid encoding PIP5KIgamma (PIP5K1c) and anti-PIP5KIgamma antibody. We thank Marchesa Bilio for expert technical assistance; we thank also IGB FACS, IGB Microscopy, and IGB Mouse Facilities for technical support. This work was funded by the Ministry of University and Research (MUR), National Recovery and Resilience Plan (2022), Prin-PNRR 2022 project P2022ZXAJ9 and Prin 2022 project 2022JAEY4L to D Alfano, National Institutes of Health R35 GM131709 to CE Turner and by Telethon Fundation project GMR22T1012 to A Baldini.

### Author Contributions

O Iacolare: data curation and methodology.
R Ferrentino: methodology.
A Altomonte: methodology.
CE Turner: funding acquisition, investigation, and writing—review and editing.
A Baldini: funding acquisition, visualization, and writing—review and editing.
D Alfano: conceptualization, data curation, supervision, funding acquisition, project administration, and writing—original draft, review, and editing.

### Conflict of Interest Statement

The authors declare that they have no conflict of interest.

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
