## [Reviewer comments · Life Science Alliance]

Tbx1 plays a critical role in focal adhesion dynamics through Paxillin regulation.

Olimpia Iacolare, Rosa Ferrentino, Alessandra Altomonte, Christopher Turner, Antonio Baldini, and Daniela Alfano
DOI: <https://doi.org/10.26508/lsa.202403151>

Corresponding author(s): Daniela Alfano, Institute of Genetics and Biophysics

Review Timeline:	Submission Date:	2024-11-28
	Editorial Decision:	2025-01-03
	Revision Received:	2025-04-24
	Editorial Decision:	2025-05-16
	Revision Received:	2025-05-20
	Accepted:	2025-05-22

Scientific Editor: Tim Fessenden

Transaction Report:

January 3, 2025

Re: Life Science Alliance manuscript #LSA-2024-03151

Dr. Daniela Alfano
Institute of Genetics and Biophysics
National Research Council
Via Pietro Castellino
Naples 80121
Italy

Dear Dr. Alfano,

Thank you for submitting your manuscript entitled "Tbx1 plays a critical role in focal adhesion dynamics through Paxillin regulation." to Life Science Alliance. The manuscript was assessed by expert reviewers, whose comments are appended to this letter. We invite you to submit a revised manuscript addressing the Reviewer comments.

Thank you for this interesting contribution to Life Science Alliance. We are looking forward to receiving your revised manuscript.

Sincerely,

B. MANUSCRIPT ORGANIZATION AND FORMATTING:

Reviewer #1 (Comments to the Authors (Required)):

localare et al have identified a role for the T-box transcription factor TBX1 in regulating focal adhesion turnover via the adaptor protein paxillin. They build on previous in vivo work using a cell culture model that permits manipulation of TBX1 and paxillin expression levels and assessment of focal adhesion dynamics, signalling pathways and integrin surface levels. This model allows them to describe a role for TBX1 in regulating focal adhesion disassembly and they make a number of interesting observations in this manuscript that are well supported by the data presented. There are some aspects of the study though that could be improved or clarified, and I think that addressing the following comments would help with this.

1. The live-cell analysis of adhesion turnover is very good and strongly supports a role for TBX1 in adhesion disassembly. For measuring the number of unstable adhesions, is this number per cell? This might be better presented as a proportion of total adhesions as the authors have stated that TBX1 knockdown leads to a reduction in total adhesion numbers (2019 paper), therefore is the reduction in unstable adhesions just due to a reduction in adhesions? Furthermore, how do the authors envisage their turnover data fitting with their previous observations on adhesion number and size? You might hypothesise that a defect in disassembly would lead to an increase in adhesion area. Perhaps an indication of the adhesion area in the cells used for this analysis might help to clarify, do they still see the reduction in adhesions following TBX1 knockdown in cells expressing GFP-vinculin or if endogenous vinculin is used as an adhesion marker for example?

2. Could the authors clarify their statement on the role of Rac activity in focal adhesion turnover? They observed an increase in Rac activity following TBX1 knockdown which they say is consistent with a role in adhesion turnover, but I think Rac activation has been shown to promote adhesion disassembly downstream of paxillin phosphorylation. I'm happy to be corrected on this though, a little more detail would help to clarify this.

3. The Western blots in this paper are generally supportive of the conclusions made, with some clear changes in adhesion signalling pathways being observed. However, an indication of consistency would be beneficial. This could either be in the form of quantification (as done in Figure 3) or by adding additional examples of blots to supplementary data. At the minute it is not clear how many times any given observation was made.

4. On a related note, some of the conclusions from the Westerns are not discussed in the text, despite them being interesting observations. For example, in Figure 1E it looks as if total and phospho-ERK levels are decreased. ERK has been linked to the regulation of adhesion turnover so this observation could well be important. Similarly, it looks as if P-JNK levels change in Figure 2C but this is not really mentioned in the results. Figure 2C also requires a blot to demonstrate re-expression of TBX1.

5. Figure 2D is a little difficult to interpret, especially the P-Cofilin levels, as the bands are not very distinct and no total levels for the proteins are presented. The authors have shown that TBX1 knockdown reduces paxillin levels in the knockdown cells but does this occur in the MEFs and do JNK and cofilin also change? Having these controls would help greatly with interpreting the data presented.

6. The experiments introducing exogenous paxillin to the TBX1 knockdowns are great and clearly demonstrate the importance of paxillin in mediating the effect of TBX1 on spreading and Fak activation, especially as increasing paxillin in the control cells doesn't influence cell spreading etc. What would have been fantastic at this point, and would really help to support the overall focus of the paper, would have been to observe a rescue of the adhesion disassembly defect seen using GFP-Vinculin as a marker of adhesion complexes as in Figure 1 (or just a rescue of adhesion size etc in fixed cells to show it is all paxillin dependent).

7. In figure 3C, despite a nice increase in PY397 FAK being observed following expression of paxillin in TBX1 knockout cells there is no decrease observed in the non-transfected knockdown cells. This is inconsistent with what is presented in Figure 1, could the authors comment on this or clarify?

8. The changes in surface levels of integrin beta1 are very interesting, especially as the surface levels increase whereas the total levels decrease. This is certainly indicative of a trafficking defect as suggested by the authors and this supported well by the PIP5K1c expression study (although in my version this is in Supplementary Figure 2 and not Figure 5 as described in the

legends). Some Western blots demonstrating PIP5K1c levels in this experiment would be beneficial.

9. Could the authors please clarify which Beta1 integrin antibody was used for the Flow Cytometry? This data very nicely supports the increase observed with the surface biotinylation approach but it is unclear whether they are looking at active or total integrin.

10. The figure legend for Figure 4 needs rewriting slightly as it doesn't reflect what is currently shown in A and B.

Reviewer #2 (Comments to the Authors (Required)):

In this manuscript, Iacolare et al build on their recent novel finding that the transcription factor TBX1 affects focal adhesions and cell ECM attachment by providing more molecular mechanism in vitro. In particular, the authors find that Tbx1 regulates FA dynamics specifically by perturbing adhesion disassembly and through paxillin signalling and this effects integrin trafficking. I believe that this mechanistic insight presented in this manuscript is interesting and potentially relevant to the Life Science Alliance journal, however some further experiments and significantly more quantification of the current results are required before the manuscript is suitable for publication.

Major comments:

Figure 1:

Is the data shown in A for FA assembly rate significant? If not you should clearly indicate this on the graph as n.s. (or state the p value).

The disassembly rate being significantly lower with Tbx kd is relatively convincing, however with the set up of the experiment it is difficult to definitively discern whether this is due to adhesion dynamics or migration rates (as FA would naturally disassemble more if the cells are migrating quicker). The evidence to support the key conclusion that Tbx1 kd specifically affects FA disassembly rate would be greatly improved by additionally using FRAP to accurately quantify FA turnover rates. This would strengthen the conclusions made about FA turnover based on the FAK blot.

For the Rac1 analysis in D, the changes in Rac1 activity are not massively obvious on the blots. The blots should be repeated 3 times and the fold change quantified (normalised to total Rac1). I do not think that total Rac1 activity in cells is particularly useful as for such Rho GTPases it's the localised activity that really impacts on cell function so ideally some imaging of Rac1 activity (e.g. FRET of a biosensor) would greatly strengthen the finding that Rac1 activity is increased upon Tbx1 kd by determining whether this is increased e.g. at the leading edge to produce a larger lamellipodium. I understand this might be a time consuming experiment so if it's not possible you should at least include in the discussion the importance of localised Rac1 activity.

Overall I think the picture of the role of Tbx1 in adhesions and Rac1 activity would be much clearer if you performed some basic cell migration timelapse experiments to definitively conclude that the altered adhesions or Rac1 activity significantly impact migration speeds of the same cells in vitro.

Like all blots throughout the manuscript, the ERK/FAK blot should be repeated 3 times and the fold changes quantified after normalisation to the equivalent total protein. Also can you comment why the loading (notably of the Erk1/2 but also the Fak and GAPDH) seems much lower in the Tbx1 kd FN condition than the equivalent control? Is this just experimental variability or a real effect (indicating that Tbx1 kd has an effect on proliferation/growth at least in FN conditions).

Figure 2:

The finding that Tbx1 positively regulates paxillin signalling is relatively convincing and backed up with some imaging and blots, however again needs quantification across at least 3 repeats. For the IF in A, the average p-pxn staining intensity per cell (or per adhesion) should be quantified across at least 10 cells per 3 repeats, as well as the size of the adhesion and number following thresholding of the image. It would also be helpful to stain for total pxn as it seems from the blot in B that both total pxn and p-pxn levels are affected by tbx1 kd. IF imaging as in A suffers from the lack of inherent normalisation (as the results may just be because the staining was 'worse' in the tbx1 kd condition compared to control) therefore repeats are vital.

The blot in B should be repeated and quantified. As in Figure 1 it seems that tbx1 kd is having an effect on loading (GAPDH and especially Pxn) so is it causing less translation of the total pxn protein or is it affecting cell viability?

The blots in C and D should also be quantified over 3 repeats as the tbx1 kd effect on p-JNK is not very obvious by eye (especially in C). For the MEFs in D you would need total pxn, total JNK and total cofilin blots to accurately determine if the tbx1 kd effect in MEFs is via phospho signalling or total protein activity. Also why was the cofilin only done in MEFs and not the C2C12 line?

Figure 3:

The rescue by exogenous pxn is very convincing and hugely strengthens the manuscript. The only comment I have on this figure is why is the GAPDH loading so different in the Erk part of C while in the Fak part the loading is very consistent?

Figure 4:

The link to integrin B1 trafficking is a nice way to round off and finish the paper and is mostly convincing, however the pull down blot should be quantified (and repeated) in A, a statistical test of significance should be added to B, and an attempt made to

quantify the staining intensity (and localisation of staining) in D.

Minor comment:

There is a Figure 5 legend however no figure 5 so this legend should be removed (it seems like the figure is S2 instead).

Reviewer #3 (Comments to the Authors (Required)):

This study aims to investigate, in vitro, the molecular and functional effects of Tbx1 on focal adhesion turnover and dynamics. The Tbx1 is implicated in heart development and several human and murine studies, cited by the authors in their introduction, suggest the in vivo relevance of Tbx1. Furthermore, gene expression data indicate that the focal adhesion pathway signaling is significantly altered upon loss of Tbx1 in vivo. However, molecular level understanding of this pathway and how Tbx1 regulates focal adhesions is currently lacking. Therefore, this cell biological study is a timely and addresses an important knowledge gap with potentially clinical and translational relevance. However, the manuscript, in its current form lacks scientific robustness and more experimental work, in particular, biologically independent repetitions of experiments, additional controls and rigorous statistical analyses are needed. In addition, it seems that loss-of-function experiments lack necessary off-target controls (independent siRNAs or rescue).

Figure 1 Please provide details for the FA analyses. How many cells, from how many independent biological repeats? What are the data points representing? For D and E, these experiments need to be repeated at least 3 times and quantification and statistics provided.

Figure 2. The same concerns as for Figure 2. For A, please provide image analysis based quantification of p-pax and indication of how many cells, from how many independent biological repeats? What are the data points representing? For all the WB data; these experiments need to be repeated at least 3 times and quantification and statistics provided.

Figure 4. The figure legend does not seem to match with the figure. The legend indicated that A is the cytosolic b1-integrin, however the images show the biotin pulldown and B perhaps the quantification.

Please clarify and provide details of what the data in B represent, how many expts (show individual data points). The methods indicate that these were blotted also for b3 but that data seems to be lacking in the figure. Please include b3-integrin in the Figure 4 data to show if this is a specific b1-integrin effect.

For the 9EG7 FACS please show histograms and include also a cell surface FACS with a total b1-integrin antibody. Is the altered cell surface levels specific to the activation status also does it reflect alterations in subcellular total b1-integrin pools. Also if the authors stimulate the TBx1 lacking cells with manganese, does this rescue the defect? This would provide mechanistic insight to whether the defect is in inside-out or outside in activation. Are talin or kindlin levels altered?

4D please provide quantification of the staining intensities from multiple samples.

Reviewer #1 (Comments to the Authors (Required)):

localare et al have identified a role for the T-box transcription factor TBX1 in regulating focal adhesion turnover via the adaptor protein paxillin. They build on previous in vivo work using a cell culture model that permits manipulation of TBX1 and paxillin expression levels and assessment of focal adhesion dynamics, signalling pathways and integrin surface levels. This model allows them to describe a role for TBX1 in regulating focal adhesion disassembly and they make a number of interesting observations in this manuscript that are well supported by the data presented. There are some aspects of the study though that could be improved or clarified, and I think that addressing the following comments would help with this.

1. The live-cell analysis of adhesion turnover is very good and strongly supports a role for TBX1 in adhesion disassembly. For measuring the number of unstable adhesions, is this number per cell? This might be better presented as a proportion of total adhesions as the authors have stated that TBX1 knockdown leads to a reduction in total adhesion numbers (2019 paper), therefore is the reduction in unstable adhesions just due to a reduction in adhesions? Furthermore, how do the authors envisage their turnover data fitting with their previous observations on adhesion number and size? You might hypothesise that a defect in disassembly would lead to an increase in adhesion area. Perhaps an indication of the adhesion area in the cells used for this analysis might help to clarify, do they still see the reduction in adhesions following TBX1 knockdown in cells expressing GFP-vinculin or if endogenous vinculin is used as an adhesion marker for example?

We appreciate the reviewer's comment. Regarding the FA turnover analysis, the number of unstable adhesions is per cell (we changed the graph labeling of Fig.1C) and the values have been already normalized with the total adhesion numbers. Furthermore, in Alfano et al, HMG 2019 we already quantify the FA area by using anti-PXN, anti-P-PXN and anti-Vcl antibodies (Fig. 5B-5C) and we still found a reduction in adhesions following Tbx1 knockdown. As shown in Deakin and Turner MBC 2012 it seems there is a complex relationship with paxillin affecting short lived and long lived adhesions.

2. Could the authors clarify their statement on the role of Rac activity in focal adhesion turnover? They observed an increase in Rac activity following TBX1 knockdown which they say is consistent with a role in adhesion turnover, but I think Rac activation has been shown to promote adhesion disassembly downstream of paxillin phosphorylation. I'm happy to be corrected on this though, a little more detail would help to clarify this. We appreciate the reviewer's comment and agree that our statement regarding the role of Rac activity in focal adhesion (FA) turnover can be clarified. Based on current evidence, we do not believe there is a clear or direct relationship between Rac1 activity and FA turnover. In our study, the observed increase in Rac1 does not necessarily imply more stable adhesions, and we acknowledge that steady-state Rac1 levels are difficult to interpret in this context.

The literature presents a complex and somewhat contradictory view on this topic. Some studies suggest that Rac is not essential for FA dynamics but may influence the dynamics of individual adhesion proteins (e.g., Steffen et al., JCS, 2013). Webb et al. (2004) proposed that paxillin phosphorylation is required for FA disassembly, though they did not identify a role for Rac1 specifically. Moreover, Deakin et al. (PLOS One, 2012) highlighted the intricate interplay between Rac/Rho signaling and the paxillin–vinculin association.

In light of this, we have revised the text in the Discussion section (line 419) to clarify our interpretation and have softened the conclusions accordingly.

3. The Western blots in this paper are generally supportive of the conclusions made, with some clear changes in adhesion signalling pathways being observed. However, an indication of consistency would be beneficial. This could either be in the form of quantification (as done in Figure 3) or by adding additional examples of blots to supplementary data. At the minute it is not clear how many times any given observation was made.

We apologize for the inconsistencies in the presentation of Western blot data, particularly regarding the quantification. In the revised manuscript, we have added the corresponding quantification graphs for all Western blot analyses to improve data clarity and presentation. All Western blot experiments were performed at least three times, and we now include quantification and representative graphs for each in the relevant figures. We hope these updates address the reviewer's concerns and enhance the overall clarity and rigor of the data.

4. On a related note, some of the conclusions from the Westerns are not discussed in the text, despite them being interesting observations. For example, in Figure 1E it looks as if total and phospho-ERK levels are decreased. ERK has been linked to the regulation of adhesion turnover so this observation could well be important. Similarly, it looks as if P-JNK levels change in Figure 2C but this is not really mentioned in the results. Figure 2C also requires a blot to demonstrate re-expression of TBX1.

We appreciate the reviewer's comment. We would like to clarify that the observed decrease in ERK levels in Tbx1 knockdown (KD) cells is likely due to experimental variability and does not reflect a real biological effect. To address this, we have replaced the original Western blot images with more representative ones. Additionally, we performed quantitative analysis and have added the corresponding graph in Figure 1E (right panel). Regarding the P-JNK levels shown in Figure 2C, we have added a clarifying comment in the Results section (line 323). With respect to TBX1 re-expression, we apologize for not including the corresponding Western blot in the original submission. We have now added this data in Supplementary Figure 1A, which shows both the knockdown and overexpression of TBX1.

5. Figure 2D is a little difficult to interpret, especially the P-Cofilin levels, as the bands are not very distinct and no total levels for the proteins are presented. The authors have shown that TBX1 knockdown reduces paxillin levels in the knockdown cells but does this occur in the MEFs and do JNK and cofilin also change? Having these controls would help greatly with interpreting the data presented.

We apologize for the quality of the P-cofilin Western blot. Unfortunately, many MEF cell clones were lost during the course of the study, making it unfeasible to perform additional experiments using MEF cells. Since the data obtained from MEF cells do not provide additional insights beyond those observed in the C2C12 cellular system, we believe it is appropriate to remove these results from the manuscript. Specifically, we have removed panel D from Figure 2, along with the corresponding text in the Results section.

6. The experiments introducing exogenous paxillin to the TBX1 knockdowns are great and clearly demonstrate the importance of paxillin in mediating the effect of TBX1 on spreading and Fak activation, especially as increasing paxillin in the control cells doesn't influence cell spreading etc. What would have been fantastic at this point, and would really help to support the overall focus of the paper, would have been to observe a rescue of the adhesion disassembly defect seen using GFP-Vinculin as a marker of adhesion complexes as in Figure 1 (or just a rescue of adhesion size etc in fixed cells to show it is all paxillin dependent).

We apologize, but we were unable to add new results to directly address this point. Unfortunately, we are no longer able to perform the adhesion disassembly assay using spinning confocal time-lapse imaging. Moreover, based on the analysis of the individual frames, we cannot draw conclusions regarding the adhesion size, as the mRFP-PXN overexpression experiment rescued the cell spreading (20 minutes after plating) rather than adhesion size in cultured cells, so the experimental conditions would be very different.

7. In figure 3C, despite a nice increase in PY397 FAK being observed following expression of paxillin in TBX1 knockout cells there is no decrease observed in the non-transfected knockdown cells. This is inconsistent with what is presented in Figure 1, could the authors comment on this or clarify?

We thank the reviewer for the opportunity to clarify that there is no inconsistency between Figures 1D and 3C. The quantification we have added supports this point. Specifically, total FAK levels are higher in TBX1 knockdown cells compared to control cells; therefore, P-FAK levels were normalized to total FAK. As shown in the graph, once normalized, P-FAK levels are indeed lower in TBX1 knockdown cells compared to control cells (which do not overexpress PXN), as reflected in the quantitative analysis.

8. The changes in surface levels of integrin beta1 are very interesting, especially as the surface levels increase whereas the total levels decrease. This is certainly indicative of a trafficking defect as suggested by the authors and this supported well by the PIP5K1c expression study (although in my version this is in Supplementary Figure 2 and not Figure 5 as described in the legends). Some Western blots demonstrating PIP5K1c levels in this experiment would be beneficial.

We appreciate the reviewer's comment regarding the increase in surface integrin β 1 levels in Tbx1 knockdown (KD) cells, which may indicate a trafficking defect, potentially due to PIP5K1c downregulation. To address this, we have added a Western blot showing PIP5K1c protein levels along with the corresponding quantitative analysis in Supplementary Figure 2B. Additionally, we have revised the text in the manuscript accordingly (line 390).

9. Could the authors please clarify which Beta1 integrin antibody was used for the Flow Cytometry? This data very nicely supports the increase observed with the surface biotinylation approach but it is unclear whether they are looking at active or total integrin.

We performed FACS analysis using both antibodies recognizing total integrin and active integrin, and the results were very similar. Therefore, we chose to carry out the entire analysis using only the anti-total integrin antibody.

10. The figure legend for Figure 4 needs rewriting slightly as it doesn't reflect what is currently shown in A and B.

We rewrote that.

Reviewer #2 (Comments to the Authors (Required)):

In this manuscript, Iacolare et al build on their recent novel finding that the transcription factor TBX1 affects focal adhesions and cell ECM attachment by providing more molecular mechanism in vitro. In particular, the authors find that Tbx1 regulates FA dynamics specifically by perturbing adhesion disassembly and through paxillin signalling and this effects integrin trafficking. I believe that this mechanistic insight presented in this manuscript is interesting and potentially relevant to the Life Science Alliance journal, however some further experiments and significantly more quantification of the current results are required before the manuscript is suitable for publication.

Major comments:

Figure 1:

Is the data shown in A for FA assembly rate significant? If not you should clearly indicate this on the graph as n.s. (or state the p value).

The disassembly rate being significantly lower with Tbx kd is relatively convincing, however with the set up of the experiment it is difficult to definitively discern whether this is due to adhesion dynamics or migration rates (as FA would naturally disassemble more if the cells are migrating quicker). The evidence to support the key conclusion that Tbx1 kd specifically affects FA disassembly rate would be greatly improved by additionally using FRAP to accurately quantify FA turnover rates. This would strengthen the conclusions made about FA turnover based on the FAK blot.

We apologize for not clearly indicating the statistical significance in the graph presented in Figure 1A. We have now clarified that the FA assembly rate is not significantly different.

We thank the reviewer for suggesting FRAP as a method to assess focal adhesion (FA) turnover, as it indeed offers valuable insights into paxillin mobility. However, we believe that measuring the time from initiation to disappearance of FAs remains the gold standard for evaluating FA lifetime. Additionally, analyzing assembly and disassembly rates provides a more direct assessment of FA dynamics, as supported by numerous publications.

While FRAP can provide important information about the dynamics of individual FA components, it does not directly reflect overall FA lifetime or turnover. We also note that our response to Reviewer 1 may address similar concerns related to this topic.

For the Rac1 analysis in D, the changes in Rac1 activity are not massively obvious on the blots. The blots should be repeated 3 times and the fold change quantified (normalised to total Rac1). I do not think that total Rac1 activity in cells is particularly useful as for such Rho GTPases it's the localised activity that really impacts on cell function so ideally some imaging of Rac1 activity (e.g. FRET of a biosensor) would greatly strengthen the finding that Rac1 activity is increased upon Tbx1 kd by determining whether this is increased e.g. at the leading edge to produce a larger lamellipodium. I understand this might be a time consuming experiment so if it's not possible you should at least include in the discussion the importance of localised Rac1 activity.

We agree with the reviewer that the localized activity of Rho GTPases is more critical for cell function than total active levels. Ideally, spatial and dynamic analyses of Rac1 activity—using techniques such as FRET or biosensors—would provide valuable insights. However, as the reviewer noted, these approaches are time-consuming and may not always fully resolve the complexity of Rac1 regulation. Nevertheless, we have acknowledged the importance of spatial dynamics of Rac activation in the Discussion section (line 419).

Overall I think the picture of the role of Tbx1 in adhesions and Rac1 activity would be much clearer if you performed some basic cell migration timelapse experiments to definitely conclude that the altered adhesions or Rac1 activity significantly impact migration speeds of the same cells in vitro.

We thank the reviewer for this comment. Unfortunately, at this time we are unable to perform time-lapse experiments at the imaging facility. However, in our previous study (Alfano et al., *Human Molecular Genetics*, 2019, Figure 4), we already demonstrated through time-lapse analysis that the migration of Tbx1KD cells is impaired. We believe these findings remain relevant and support the conclusions of the current work.

Like all blots throughout the manuscript, the ERK/FAK blot should be repeated 3 times and the fold changes quantified after normalisation to the equivalent total protein. Also can you comment why the loading (notably of the Erk1/2 but also the Fak and GAPDH) seems much lower in the Tbx1 kd FN condition than the equivalent control? Is this just

experimental variability or a real effect (indicating that Tbx1 kd has an effect on proliferation/growth at least in FN conditions).

We appreciate the reviewer's comment. It is well established that Tbx1 knockdown does not affect cell proliferation or growth. To support this, we have included a proliferation graph in Supplementary Figure 2A, showing data collected at three different time points following siRNA transfection. Furthermore, cells were plated onto fibronectin (FN) for only 20 minutes—a very short time period that is unlikely to influence the proliferation rate.

Figure 2:

The finding that Tbx1 positively regulates paxillin signalling is relatively convincing and backed up with some imaging and blots, however again needs quantification across at least 3 repeats. For the IF in A, the average p-pxn staining intensity per cell (or per adhesion) should be quantified across at least 10 cells per 3 repeats, as well as the size of the adhesion and number following thresholding of the image. It would also be helpful to stain for total pxn as it seems from the blot in B that both total pxn and p-pxn levels are affected by tbx1 kd. IF imaging as in A suffers from the lack of inherent normalisation (as the results may just be because the staining was 'worse' in the tbx1 kd condition compared to control) therefore repeats are vital.

We appreciate the reviewer's insightful comment. In Figure 2A, we have added the quantification (in terms of size and number) of P-Pxn- and Tot-Pxn- based FA staining, based on immunofluorescence (IF) experiments performed on 30 cells across three independent repeats. We reported also the ratio P-Pxn/Pxn to show the decreased of Pxn phosphorylation in Tbx1KD cells. Unfortunately, we could not use the same cells for both P-Pxn and total Pxn staining, as the antibodies used are both rabbit polyclonal. Additionally, we are aware that total Pxn levels are reduced in Tbx1 knockdown cells. As previously demonstrated in Alfano et al., 2019, Tbx1 transcriptionally regulates Paxillin. Therefore, the decrease in both total and phosphorylated Pxn upon Tbx1 knockdown is consistent with our earlier findings. We revised the text accordingly (line 298).

The blot in B should be repeated and quantified. As in Figure 1 it seems that tbx1 kd is having an effect on loading (GAPDH and especially Pxn) so is it causing less translation of the total pxn protein or is it affecting cell viability?

We present a quantitative graph summarizing results from three independent experiments. It is well established that Paxillin (Pxn) is a transcriptional target of Tbx1, as previously demonstrated in Alfano et al., *Human Molecular Genetics*, 2019, and Fulcoli et al., *Nature Communications*, 2018. As also stated in the manuscript (line 330), we therefore expected a decrease in Pxn levels in Tbx1 knockdown (KD) cells. As previously mentioned, Tbx1 KD does not affect cell viability, at least within 48 hours post-siRNA transfection, as shown in Supplementary Figure 2A.

The blots in C and D should also be quantified over 3 repeats as the tbx1 kd effect on p-JNK is not very obvious by eye (especially in C). For the MEFs in D you would need total pxn, total JNK and total cofilin blots to accurately determine if the tbx1 kd effect in MEFs is via phospho signalling or total protein activity. Also why was the cofilin only done in MEFs and not the C2C12 line?

Unfortunately, many MEF cell clones were lost during the course of the study. Since the data obtained from MEF cells do not provide additional insights beyond what was observed in the C2C12 cellular system, we have decided to remove these data from the manuscript. Regarding the analysis of phosphorylated JNK (p-JNK), we performed three or more independent experiments and quantified the results, as shown in the graph in Figure 2C (right panels).

Figure 3:

The rescue by exogenous pxn is very convincing and hugely strengthens the manuscript. The only comment I have on this figure is why is the GAPDH loading so different in the Erk part of C while in the Fak part the loading is very consistent?

The two blots shown are from separate loading experiments. However, the quantification was normalized to GAPDH levels to ensure accuracy and comparability between samples.

Figure 4:

The link to integrin B1 trafficking is a nice way to round off and finish the paper and is mostly convincing, however the pull down blot should be quantified (and repeated) in A, a statistical test of significance should be added to B, and an attempt made to quantify the staining intensity (and localisation of staining) in D.

The pull-down quantification (N=3 expts) was shown in (B). We also quantified the staining intensity in D. The data were added in the Fig. 4E (lower panel).

Minor comment:

There is a Figure 5 legend however no figure 5 so this legend should be removed (it seems like the figure is S2 instead).

We removed figure 5 legend.

Reviewer #3 (Comments to the Authors (Required)):

This study aims to investigate, in vitro, the molecular and functional effects of Tbx1 on focal adhesion turnover and dynamics. The Tbx1 is implicated in heart development and several human and murine studies, cited by the authors in their introduction, suggest the in vivo relevance of Tbx1. Furthermore, gene expression data indicate that the focal adhesion pathway signaling is significantly altered upon loss of Tbx1 in vivo. However, molecular level understanding of this pathway and how Tbx1 regulates focal adhesions is currently lacking. Therefore, this cell biological study is a timely and addresses an important knowledge gap with potentially clinical and translational relevance. However, the manuscript, in its current form lacks scientific robustness and more experimental work, in particular, biologically independent repetitions of experiments, additional controls and rigorous statistical analyses are needed. In addition, it seems that loss-of-function experiments lack necessary off-target controls (independent siRNAs or rescue).

We used a pool of three different siRNAs targeting distinct sequences of the gene. The results obtained using the siRNAs either in combination or individually were consistent, allowing us to exclude off-target effects. Furthermore, all experiments were performed using biologically independent samples and were reproducible. For each experiment, we collected at least three biological replicates.

Figure 1 Please provide details for the FA analyses. How many cells, from how many independent biological repeats? What are the data points representing? For D and E, these experiments need to be repeated at least 3 times and quantification and statistics provided.

We thank the reviewer for this comment.

In the Supplementray Fig.1 C, we showed dot plot FA analysis by measuring 70 cells. We did the experiments at least 3 times and provided relative quantification and statistics in the graphs.

Figure 2. The same concerns as for Figure 2. For A, please provide image analysis based quantification of p-pax and indication of how many cells, from how many independent biological repeats? What are the data points representing? For all the WB data; these experiments need to be repeated at least 3 times and quantification and statistics provided.

We did the experiments at least 3 times and provided relative quantification in graph. We provided P-Pxn and Tot-Pxn staining quantifications, as well as the quantification derived from P-Pxn/Tot-Pxn ratio.

Figure 4. The figure legend does not seem to match with the figure. The legend indicated that A is the cytosolic b1-integrin, however the images shows the biotin pulldown and B perhaps the quantification. Please clarify and provide details of what the data in B represent, how many exps (show individual data points). The methods indicate that these were blotted also for b3 but that data seems to be lacking in the figure. Please include b3-integrin in the Figure 4 data to show if this is a specific b1-integrin effect.

We apologize for the oversight and have corrected the figure legend to accurately reflect the data shown (surface integrin, not cytosolic fraction). We performed the biotin pull-down experiment three times and have included the corresponding quantification. However, we would like to emphasize that the most robust and statistically significant data regarding integrin surface levels are derived from the FACS analysis presented in Figure 4C.

We have also revised the Methods section accordingly, removing the reference to blotting with β 3 integrin.

As the reviewer rightly pointed out, we cannot conclusively determine whether the observed effect is specific to β 1 integrin, given that PIP5K1c is known to regulate the trafficking of multiple receptors. It is therefore plausible that other integrin subunits may also be affected. We have acknowledged this limitation and clarified this point in the Results section (lines 388-392).

For the 9EG7 FACS please show histograms and include also a cell surface FACS with a total b1-integrin antibody. Is the altered cells surface levels specific to the activation status also does it reflect alterations in subcellular total b1-integrin pools.

We thank the reviewer for the opportunity to clarify this point. We previously performed FACS analysis using three different antibodies: two recognizing total β 1 integrin and one specific for the active form (9EG7). For consistency, we have presented the results obtained with one representative antibody in the current manuscript. Overall, we believe that the observed changes in active β 1 integrin levels reflect corresponding changes in total β 1 levels.

Also if the authors stimulate the Tbx1 lacking cells with manganese, does this rescue the defect? This would provide mechanistic insight to whether the defect is in inside-out or outside in activation. Are talin or kindlin levels altered?

Thank you for raising this important point, we appreciate this comment. We performed Western blot analysis using a specific antibody against Talin and observed a slight increase in Talin expression levels in Tbx1 knockdown (KD) cells compared to control cells, although this difference is not statistically significant. We have included this result in Supplementary Figure 2C and updated the corresponding text in the Results section (line 395).

4D please provide quantification of the staining intensities from multiple samples.

We included the quantification in the Figure 4D (lower panel).

May 16, 2025

RE: Life Science Alliance Manuscript #LSA-2024-03151R

Dr. Daniela Alfano
Institute of Genetics and Biophysics
National Research Council
Via Pietro Castellino
Naples 80131
Italy

Dear Dr. Alfano,

Thank you for submitting your revised manuscript entitled "Tbx1 plays a critical role in focal adhesion dynamics through Paxillin regulation." As you will see, reviewers are overall satisfied with the changes made in this revision. We invite you to consider the final note by Reviewer 3 on measuring focal adhesion proteins via FACS following Mg stimulation. Resolving this point with additional data is left to your discretion. We would be happy to publish your paper in Life Science Alliance pending that final point as well as revisions necessary to meet our formatting guidelines.

- Please upload your main manuscript text as an editable .doc file.
- Please upload your main and supplementary figures as single files.
- Please add the X and Bluesky handles of your host institute/organization, as well as your own and/or one of the authors, to our system.
- Please be sure that the authorship listing and order are correct and match between the system and the manuscript file.
- The full name (middle names as initials) of each author should be given on the title page of the manuscript in the form First name, middle name, Last name.
- Please consult our manuscript preparation guidelines <https://www.life-science-alliance.org/manuscript-prep> and make sure your manuscript sections are in the correct order.
- Please add your main and supplementary figure legends to the main manuscript text after the references section.
- Please be sure that all authors are mentioned in the authors' contribution section.
- Please use the [10 author names, et al.] format in your references (i.e., limit the author names to the first 10).
- Please add a Conflict of Interest statement to your main manuscript text.
- Please add an Ethical Approval statement concerning the use of animal models.
- Please add a Data Availability statement.
- We encourage you to revise the legend for Figure 2 such that the figure panels are introduced in alphabetical order.
- Please ensure the scale bars in Figure 4 are clearly visible.
- Please add callouts for Figures 2D; 4D and S2A to your main manuscript text.
- Please add molecular weight markers for all western blots.

LSA now encourages authors to provide a 30-60 second video where the study is briefly explained. We will use these videos on social media to promote the published paper and the presenting author (for examples, see <https://docs.google.com/document/d/1-UWCfbE4pGcDdcgzcmiuJl2XMBJnxKYeqRvLLrLS08s/edit?usp=sharing>). Corresponding or first-authors are welcome to submit the video. Please submit only one video per manuscript. The video can be emailed to contact@life-science-alliance.org

A. FINAL FILES:

B. MANUSCRIPT ORGANIZATION AND FORMATTING:

Sincerely,

Reviewer #1 (Comments to the Authors (Required)):

I would like to thank the authors for addressing my comments from the initial review. I am happy with their responses and the changes they have made to the manuscript and I am happy to support the publication of this paper.

Reviewer #2 (Comments to the Authors (Required)):

I believe that the authors have satisfactorily responded to my comments/queries on the original manuscript and as a result this version is now acceptable for publication without further revisions.

Reviewer #3 (Comments to the Authors (Required)):

The authors have addressed some of the point I have raised in my review. However, it seems they may have missed one important point raised as they have not carried out the suggested experiments:

"Also if the authors stimulate the TBx1 lacking cells with manganese, does this rescue the defect? This would provide mechanistic insight to whether the defect is in inside-out or outside in activation."

Testing this is easy with the FACS system they have up and running and would significantly increase the value of the study by providing important mechanistic insight

May 22, 2025

RE: Life Science Alliance Manuscript #LSA-2024-03151RR

Dr. Daniela Alfano
Institute of Genetics and Biophysics
National Research Council
Via Pietro Castellino
Naples 80131
Italy

Dear Dr. Alfano,

Thank you for submitting your Research Article entitled "Tbx1 plays a critical role in focal adhesion dynamics through Paxillin regulation.". It is a pleasure to let you know that your manuscript is now accepted for publication in Life Science Alliance. Congratulations on this interesting work.

DISTRIBUTION OF MATERIALS:

Again, congratulations on a very nice paper. I hope you found the review process to be constructive and are pleased with how the manuscript was handled editorially. We look forward to future exciting submissions from your lab.

Sincerely,
